# ALPHA-RF: AUTOMATED RF-FILTER-CIRCUIT DESIGN WITH NEURAL SIMULATOR AND REINFORCEMENT LEARNING

## ABSTRACT

Accurate, high-performance radio-frequency (RF) filter circuits are ubiquitous in radio-frequency communication and sensing systems for accepting and rejecting signals at desired frequencies. Conventional RF filter design process involves manual calculations of design parameters, followed by intuition-guided iterations to achieve the desired response for a set of filter specifications. This process is time-consuming due to time- and resource-intensive electromagnetic simulations using full-wave numerical PDE solvers, and requires many intuition-guided adjustments to achieve an practically usable design. This process is also highly sensitive to domain expertise and requires many years of professional training. To address these bottlenecks, we propose an automatic RF filter circuit design tool using neural simulator and reinforcement learning. First, we train a neural simulator to replace the PDE electromagnetic simulator. The neural-network-based simulator reduces each of the simulation time from 4 minutes on average to less than 100 millisecond while maintaining a high precision. Such dramatic acceleration enable us to leverage deep reinforcement learning algorithm and train an amortized inference policy to perform automatic design in the imagined space from the neural simulator. The resulted automatic circuit-design agent achieves super-human design results and exceeds specifications in several cases. The automatic circuit-design agent also reduces the on-average design cycle from days to under a few seconds. Even more surprisingly, we demonstrate that the neural simulator can generalize to design spaces far from the training dataset and in a sense it has learned the underlying physics–Maxwell equations. We also demonstrate that the reinforcement learning has discovered many expert-like design intuitions. This work marks a step in using neural simulators and reinforcement learning in RF circuit design and the proposed method is generally applicable to many other design problems and domains in close affinity.

## 1 INTRODUCTION

In recent years, continuous improvement in radio-frequency (RF) circuit design, which is a subset of analog circuit design that deals with the **generation, amplification and manipulation of high-frequency signals**, has allowed for reliable, high-performance RF circuits and systems that enable 5G wireless (20), Internet-of-Things, high-speed optical communication, etc. Even though several RF system architectures (25) exist for different applications and technologies, virtually all of them employ accurate, high-performance filters to perform transmission and rejection of RF signals at desired frequencies (26; 27).

The conventional filter design process starts with calculations of design parameters given a set of specifications, which results in an initial design. Electromagnetic (EM) simulation is then performed using full-wave numerical PDE solvers to obtain the initial S-parameters, which often fails to meet specifications due to complex EM coupling at high frequencies. The engineer must then leverage intuition, iterating upon the design several times to arrive at a practically usable design. This process is often time-consuming due to the high number of time- and resource-intensive EM simulations required, and highly sensitive to domain expertise.

Given these bottlenecks, an accelerated design procedure shall consist of two key components: 1) A fast, computationally-inexpensive simulator 2) An automated designer with expertise comparable or superior to experienced RF engineers. For 1), fast neural simulators as surrogates to numerical solvers have shown considerable success in applications over many domains, such as neuroscience (29; 24), particle simulation (28; 11; 1), motion control and simulation (17; 9) and photonics (8; 2; 13; 22; 4) with acceleration up to five orders of magnitude. Recently, EM neural simulators have also been explored in RF integrated circuits (RFICs) design to rapidly predict frequency responses of RF circuits and structures, enabling automated design (15; 6). These works treat electromagnetic simulation as an image analysis problem where a neural network learns the mapping between geometrical features of a circuit and its frequency response. While implemented for chip-level circuits, this formulation is general and potentially applicable to system-level filters. Turning to 2), recent works in automated filter design have implemented deep-Q network (21) and generated neural network (18) to perform automated tuning. These methods require an initial design obtained through either equation-based surrogate modeling, which may be inaccurate at higher frequencies where EM coupling is complex, or measured real-life prototypes. Inverse design, on the other hand, does not require an initial design, as shown in an implementation using generative adversarial network (30) with a comparatively long design time of 11 minutes, and genetic algorithms (3).

In this paper, we propose **Alpha-RF**, an automatic design agent leveraging neural simulator and deep reinforcement learning (RL) with amortized inference. The acceleration comes two-fold. First, we replace the PDE solver with a high-precision neural-network-based S-parameters simulator. Trained on a dataset of filter layouts and S-parameters, a $2 \times 2$ matrix which fully describes the frequency response of a design, the neural simulator learns to rapidly predict the S-parameters of any filter design when given an image of the layout, with precision comparable to a full-wave PDE solver over a broad range of filter configurations. The neural simulator only requires **less than 100 milliseconds per prediction** compared to 4 minutes with a full-wave solver. Second, we amortize the search with RL in a two-phase pipeline, leveraging the significant acceleration from the neural simulator: during training, a spec-conditioned policy is optimized via neural simulator roll-outs. At inference, amortized inference maps target specifications to a layout **in a single forward pass**, optionally sampling a few candidates and validating with the fast simulator, providing high-quality designs while dramatically improving time efficiency. Demonstrations of Alpha-RF through six different filter specifications show comparable or superior designs to those by experienced RF engineers, all done within seconds. This performance level suggests learning of human design intuitions shown through selective targeting of key design parameters for certain specifications. Examples of a different class of circuits also show the neural simulator's prediction capability beyond filter circuits.

Our key contributions in this work are as follows: 1) Scalable, high-precision neural S-parameters simulator for RF filter circuits to replace time-consuming full-wave numerical PDE solvers, with **more than three-orders-of-magnitude acceleration** in simulation time. 2) An automatic RL design agent (Alpha-RF) leveraging the fast neural simulator to generate accurate, high-quality filter circuits comparable to and, in several cases, outperform designs by experienced human RF engineers **within seconds**. 3) A design automation framework applicable to related design problems.

## 2 METHOD

To accelerate the expensive EM-simulation-based design process, we build a neural-network-based digital twin to predict S-parameters of filter layouts. This fast, differentiable surrogate makes it feasible to run reinforcement learning, which requires millions of training steps, and allows the agent to rapidly explore and deliver designs within our simulator. In this section, we first describe the construction of our neural simulator, then present the RL environment setup, and finally introduce learning algorithm. Together, these components form Alpha-RF, our end-to-end framework for automatic RF filter design.

### 2.1 NEURAL S-PARAMETERS SIMULATOR

**Model Architecture.** The frequency response of a filter layout, which is represented by its S-parameters, is controlled by layout parameters in template $a$

$$a = (N, L, \mathrm{cw}_0, \ldots, \mathrm{cw}_7),$$

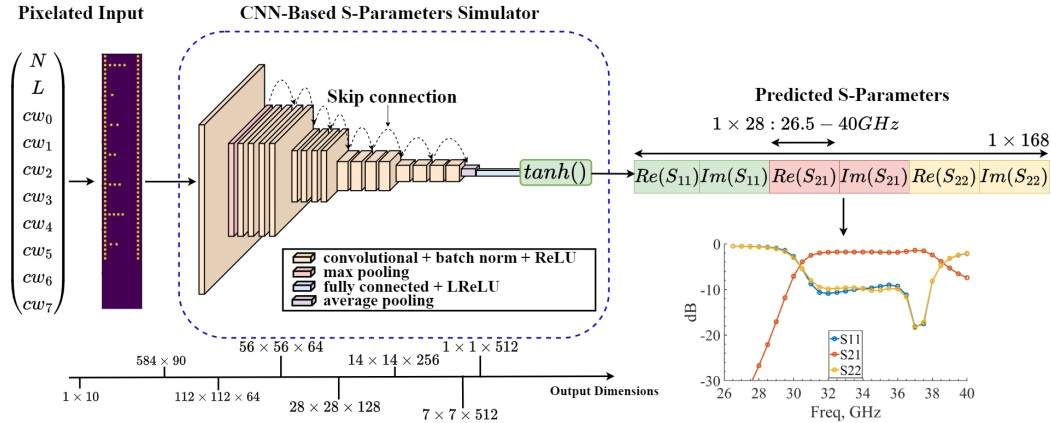

Figure 1: CNN-based S-Parameters predictor

where $N \in \{2, 3, 4, 5, 6, 7\}$ denotes the number of resonator sections (i.e filter order), $L \in [2.0, 3.6]$ mm specifies the resonator length, and $cw_0, \ldots, cw_7 \in [1.2, 2.6]$ mm represent the width of the coupling opening between adjacent resonators. The physical filter is described in details in Appendix B.1. Dimensions in $a$ are formed by rows and columns of vertical metal-metal interconnects called **'vias'**. Therefore, S-parameters are controlled by the placements of these vias. S-parameters simulation of a filter layout is thus analogous to learning the mapping of via placements to S-parameters through a convolutional neural network (CNN). Figure 1 shows the architecture of the CNN-based S-parameters simulator for filter circuits. Input to the network is a digitized two-dimensional image of the via footprint, where '1' (yellow) indicates presence of via and '0' (purple) otherwise. Output to the network is a $1 \times 168$ array with the following terms

$$S = [Re(S_{11}), Im(S_{11}, Re(S_{21}, Im(S_{21}, Re(S_{22}, Im(S_{22})]$$

where each $1 \times 28$ term consists of either the real or imaginary part of S-parameter values at a frequency across the range $26.5 - 40GHz$. The network consists of convolutional layers based on the ResNet-18 architecture (10), followed by a leaky rectified linear unit (LReLU) into a hyperbolic tangent (tanh) output layer, which bounds output range to the range of S-parameters, which is $[-1, 1]$. Because the neural simulator is evaluated by numerical accuracy compared to ground-truth from full-wave simulators, we use mean-absolute-error (MAE) loss function for training.

**Data Preparation.** The neural simulator is trained on a dataset of 100k filter layouts and their corresponding S-parameters, where the general geometry is shown in Figure 1. Ground-truth S-parameters are obtained from a full-wave electromagnetic simulator. To maintain a sufficiently large design space to cover a wide range of specifications while simplifying the search to only practical solutions, the maximum number of resonators $N$ is empirically chosen to be 7. A $N$-th order filter has up to $cw_N$ non-zero entries, while the remaining $cw$ terms, if any, are set to 0. Using $a$, via footprint of each layout is redrawn as a 1-channel image with pixel size 50 $\mu$m to resolve the smallest variation in the design space (100 $\mu$m). To accommodate for the largest physical layout, image size is chosen to be $584 \times 90$ which corresponds to a physical layout area of $29.2mm \times 4.5mm$.

**Data Scaling.** Scaling is a key driver of progress in modern AI: performance improves predictably as model size, compute, and data scale up (14). To test whether our predictor follows this trend, we perform a data-scaling ablation. As shown in Figure 3(b), increasing training data size consistently improves prediction accuracy, indicating a strong *scaling property*.

## 2.2 AUTOMATIC DESIGN WITH REINFORCEMENT LEARNING AND AMORTIZED INFERENCE

**Problem Formulation.** We formulate automatic filter design as a single-step decision-making problem, as described in Figure 2. Given target specifications, the agent outputs a complete set of design parameters in one shot. These parameters are passed through our fast neural simulator to predict the filter response, and the mismatch with the target response is converted into a scalar reward to train the policy. This amortized formulation avoids slow iterative optimization and enables efficient reinforcement learning–based design.

Figure 2: **Workflow of Alpha-RF.** During the *training phase* (left), given target specifications, the agent outputs a complete set of design parameters in one shot. These parameters are passed through the neural simulator to obtain predicted S-parameters, and the reward function converts the mismatch between measurements and target specifications into a scalar reward to update the agent. During the *inference phase* (right), the trained agent generates multiple candidate designs for target specifications, which are evaluated by the neural simulator. The candidate yielding the highest reward is selected as the final design.

**State Space $\mathcal{S}$.** In this formulation, the state corresponds directly to the target specification:

$$s = [f_0, \text{fbw}, \max S_{21}, \alpha_r, \alpha_l]$$

where $f_0$, fbw, $\max S_{21}$, $\alpha_r$, and $\alpha_l$ denote the center frequency, relative bandwidth, peak insertion loss, and stop-band rejection levels. The exact ranges of these specifications are provided in the Appendix C.1. At environment reset, target specifications are sampled from uniform distributions.

**Action Space $\mathcal{A}$.** The action directly corresponds to the circuit design parameters introduced in Section 2.1 :

$$a = (N,\ L,\ \text{cw}_0, \ldots, \text{cw}_7),$$

where $N$ is the number of resonators, $L$ is the resonator length, and $\text{cw}_i$ denote the spacings between adjacent resonators. All continuous variables are normalized to $[-1, 1]$. Thus, the action space fully specifies a candidate filter layout.

**Transition Dynamics.** Given a candidate design $a$, its predicted response is obtained through the neural S-parameters simulator introduced in Section 2.1:

$$\widehat{\mathbf{s}} = \mathcal{M}_\psi(a),$$

where $\mathcal{M}_\psi$ is the neural simulator parameterized by $\psi$. Since the task is formulated as a single-step optimization problem, the "transition" simply maps the design parameters to a resulting response state. The surrogate provides fast and high-fidelity prediction of circuit behavior without costly full-wave EM simulations. This replacement dramatically accelerates the state-transition step and thus enables the reinforcement learning for automated filter design.

**Episode Termination.** The environment consists of a single decision step: each episode terminates immediately after the agent outputs a set of design parameters.

**Reward Function.** The reward function evaluates quality of a design candidate, which is represented by its S-parameters, by comparing S-parameters measurements to specifications. These measurements are defined as

$$s_m = [f_{0m},\ fbw_m,\ \max S_{21m},\ \alpha_{rm}, \alpha_{lm}]$$

where subscript $m$ denotes measurement to differentiate from specifications in state space $s$. Because the reward function shapes the agent's exploration in the design space, we conceptualize the function as a close approximation of the human designer's judgement of design quality ("Is this a satisfactory layout for our specifications?") while allowing the possibility of specification-exceeding (superhuman) solutions. Regardless of specifications, a satisfactory filter design has the following qualities: **1)** $f_0$, $fbw$ are accurate to within 10% tolerance. An incorrect pass-band nullifies the design in its entirety **2)** $\max S_{21}$ is more than or equal to specification **3)** $\alpha_r$, $\alpha_l$ is less than or equal to

specifications. We shall incorporate these qualities into our reward function in the following ways: **1)** The reward function is the sum of sub-rewards for each specification, each sub-reward weighted by importance in real-life applications **2)** Sub-rewards for $f_0$, $fbw$ are given the highest weights to ensure the highest accuracy **3)** Sub-rewards for $\max S_{21}, \alpha_r$, $\alpha_l$ are immediately maximized when measurements exceed specifications ('superhuman' designs). The reward function $R$ is then formulated as

$$R = 0.3 r_{f_0} + 0.3 r_{fbw} + 0.2 r_{\max S_{21}} + 0.1 r_{\alpha_l} + 0.1 r_{\alpha_r}$$

where $r_{f_0}, r_{fbw}, r_{\max S_{21}}, r_{\alpha_l}, r_{\alpha_r}$ denote the sub-rewards for $f_0$, $fbw$, $\max S_{21}$, $\alpha_l$, and $\alpha_r$. Each $r$ is the ratio between measurement and specification, with formulas given in Appendix C.2.

## 2.3 REINFORCEMENT LEARNING ALGORITHM

We adopt Truncated Quantile Critics (TQC) (16) for its stability and strong continuous-control performance. A key challenge is that the first action dimension is *discrete* ($N \in \{2, 3, 4, 5, 6, 7\}$) while the remaining nine are continuous geometry parameters. A naïve scheme outputs a real $\tilde{N} \in (-1, 1)$ and quantizes it by $N = 2 + \text{round}\left(\frac{5}{2}(\tilde{N} + 1)\right)$ before simulation, which causes (i) *non-differentiability* (no gradient reaches the first-action logit/mean) and (ii) *early exploration collapse* (initial outputs cluster near zero and round to a single $N$).

To address this, we redesign the first output dimension as a Gumbel-Softmax layer (12), which samples a categorical distribution over the six possible filter orders while remaining differentiable via the straight-through estimator. This allows end-to-end gradient flow during policy optimization, making $N$ fully learnable and restoring the agent's ability to explore different filter orders.

**Hybrid Actor.** We replace the first output with a Gumbel–Softmax head and keep a squashed Gaussian head for the continuous part. Given features $h_\theta(s)$ from a residual MLP, the actor parameterizes

$$\pi_N(N \mid s) = \text{Cat}(\ell(s)), \quad \ell(s) = W_N h_\theta(s), \qquad \pi_c(a_c \mid s) = \mathcal{TN}(\mu(h_\theta(s)), \sigma(h_\theta(s))),$$

where $\mathcal{TN}$ denotes a diagonal Gaussian followed by $\tanh$ (squashing). We sample the continuous coordinates via the reparameterization trick,

$$a_c = \tanh(\mu + \sigma \odot \varepsilon), \quad \varepsilon \sim \mathcal{N}(0, I),$$

and draw a relaxed one-hot for $N$ using Gumbel–Softmax with temperature $\tau$,

$$\tilde{\mathbf{z}} = \text{softmax}((\ell + \mathbf{g})/\tau), \quad g_k \sim \text{Gumbel}(0, 1).$$

**The Objective.** We optimize the standard SAC/TQC maximum-entropy objective,

$$J_\pi(\theta) = \mathbb{E}_{s \sim \mathcal{D}, a \sim \pi_\theta} \left[ \alpha \left( \log \text{Cat}(N \mid \ell(s)) + \log \mathcal{TN}(a_c \mid \mu(s), \sigma(s)) \right) - Q(s, a) \right],$$

so gradients propagate through both the continuous reparameterization and the discrete logits via the Gumbel–Softmax path. This hybridization makes $N$ learnable, prevents early collapse to a single order, and restores exploration over filter orders while remaining end-to-end differentiable.

**Test-Time Sampling.** At inference, we leverage stochastic nature of learned policy by sampling $K$ candidate designs for each target specification. Each candidate is evaluated with a *virtual reward* predicted by the neural simulator, and the design with the highest reward is selected. In later sections, we show that this simple sampling strategy significantly improves final design quality.

## 3 RESULTS

In this section, we first validate the predictive accuracy of our neural simulator, using results from the PDE solver as ground truth, establishing a reliable foundation for subsequent reinforcement learning experiments. We then present the designs generated by our reinforcement learning agent and compare them against human expert performance. Remarkably, our method achieves designs comparable or superior to human-level designs and does so with a speed advantage of several orders of magnitude, completing in seconds what typically requires hours of manual tuning. Finally, we conduct the ablation study to isolate the contribution of Test-Time Sampling in our algorithm, demonstrating that this design choice plays a critical role in achieving the better performance. Collectively, these results highlight the effectiveness and robustness of our approach.

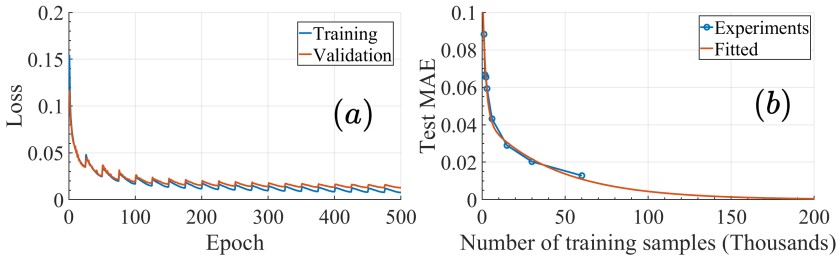

Figure 3: a) Training dynamics of the neural S-parameters simulator b) Theoretical scaling of accuracy

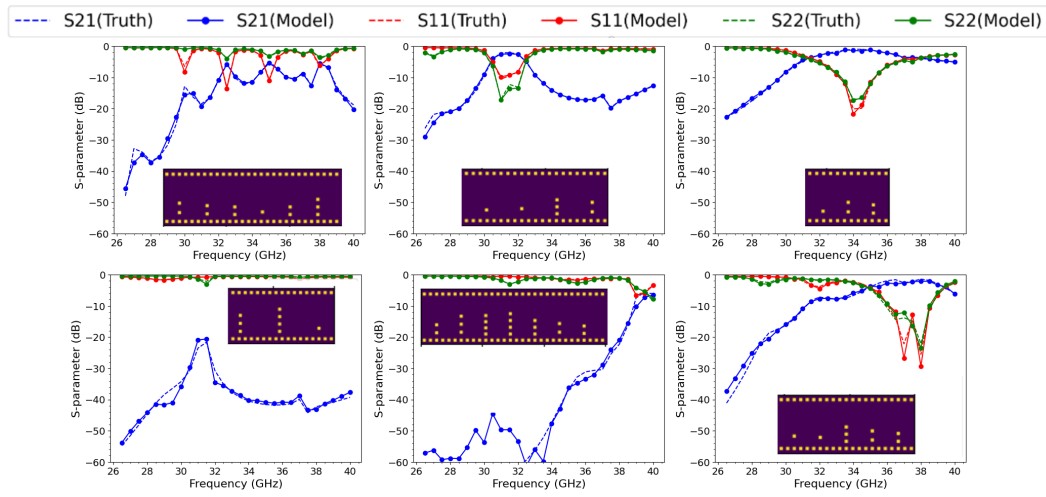

Figure 4: Comparison of predicted S-parameters and true S-parameters

## 3.1 PRECISION OF NEURAL S-PARAMETERS SIMULATOR.

In real-world applications, magnitude of S-parameters in decibel (dB) scale is of primary interest to designers. As such, we evaluate the precision of the neural simulator through average prediction uncertainty and inference examples in dB, where output from the neural simulator is compared to ground truth from a full-wave solver for different layouts.

**Inference Results.** We compare predicted S-parameters for several filter layouts from the test dataset with full-wave simulation results (ground truth) in Figure 4. It can be seen that the neural simulator accurately predicts S-parameters of a broad range of layout geometries. More specifically, even moderately complex functions and functions with consistently low values, which is more penalizing to accuracy, are accurately captured across the frequency range. Each inference takes approximately 100 milliseconds of GPU time, a more than three orders of magnitude reduction in simulation time compared to a numerical PDE solver. These results establish the CNN-based predictor as a neural equivalent to a full-wave solver for simulating the S-parameters of filter circuits.

**Prediction Uncertainty.** If $S = S_p \pm L$ where $S$ is the ground-truth S-parameters (from the PDE simulator), $S_p$ is the predicted S-parameters and $L$ is absolute error, dB difference between $S$ and $S_p$ is $20 \log_{10}(1 \pm \frac{L}{S})$, or prediction uncertainty in dB. Given that the trained neural simulator achieves a test mean-absolute-error of 0.012, average prediction uncertainty is $[1.1, 0.1]$ dB for the range of S-parameters that requires precision, $[0.1, 1]$ ($[-20, 0]$ dB). This uncertainty is well within the accepted range in real-life applications. Training dynamics of the neural simulator are shown in Figure 3(a), where the final training loss and validation loss are 0.007 and 0.012, respectively.

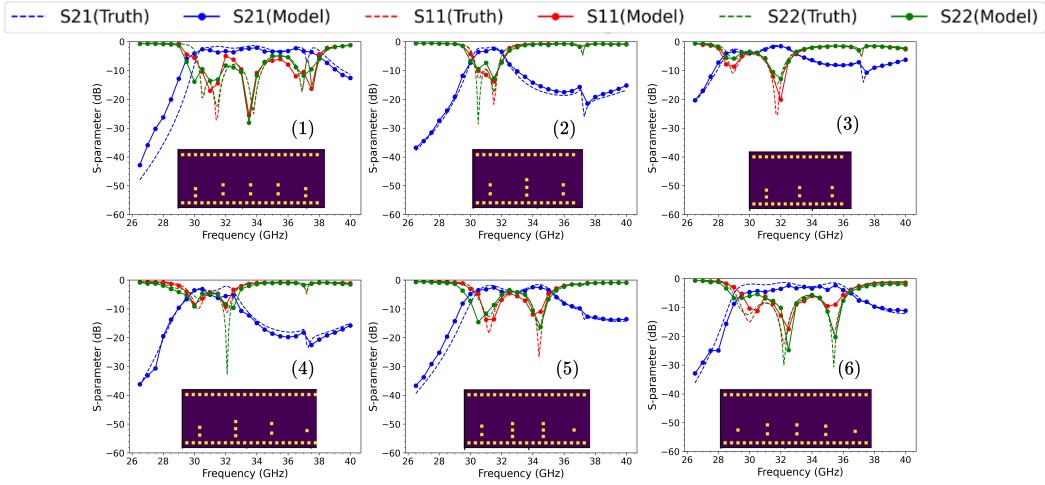

Figure 5: S-Parameters for automatically-generated filter designs of various specifications

Table 1: Specifications, model-predicted and true measurements, and rewards of designs in Figure 5.

| Design | $s$ | $s_m$ (model) $s_m$ (true) | $R$ (model) $R$ **(true)** |
|---|---|---|---|
| (1) | $[35, 0.20, -2, -20, -10]$ | $[34.35, 0.209, -1.92, -23.00, -12.92]$ $[34.25, 0.236, -1.39, -28.09, -11.33]$ | 0.9693 **0.8517** |
| (2) | $[31, 0.06, -3, -20, -10]$ | $[31.45, 0.060, -4.18, -26.67, -15.71]$ $[31.15, 0.061, -1.89, -23.00, -14.97]$ | 0.9265 **0.9783** |
| (3) | $[31, 0.15, -2, -20, -10]$ | $[30.95, 0.145, -1.94, -17.81, -8.498]$ $[30.75, 0.146, -1.27, -17.51, -7.766]$ | 0.9448 **0.9299** |
| (4) | $[31, 0.10, -3, -20, -10]$ | $[31.25, 0.099, -3.60, -22.56, -13.59]$ $[31.15, 0.099, -2.13, -19.48, -17.54]$ | 0.9476 **0.9813** |
| (5) | $[33, 0.10, -3, -20, -10]$ | $[34.10, 0.997, -3.41, -17.11, -15.78]$ $[33.55, 0.098, -1.63, -21.01, -10.57]$ | 0.9134 **0.9612** |
| (6) | $[33, 0.20, -1, -20, -10]$ | $[33.35, 0.201, -1.82, -29.06, -13.40]$ $[32.80, 0.226, -1.34, -24.62, -10.31]$ | 0.8903 **0.8484** |

## 3.2 AUTOMATIC DESIGNER

**Demonstration.** We use Alpha-RF to generate filter designs for six sets of specifications specified in Table 1. Leveraging the design tool's low latency, for each specification, the design with the highest reward is selected among **10000 candidates** generated in approximately **7 seconds**. S-parameters in dB scale for the design with the highest reward for each set is shown in Figure 5, including both neural simulator prediction and ground truth from a full-wave solver. Rewards are summarized in Table 1. All solutions have higher than 0.8903 reward for predictions (model) and more than 0.8484 for ground truth, which is lower than predictions due to error of the neural S-parameters simulator. Penalties primarily come from errors in $f_0$ and $fbw$ which are given the highest weights to ensure accurate frequency response. Nonetheless, relative errors are less than $3.30\%$ and $4.50\%$ for $f_0$ and $fbw$ for prediction results, respectively, which are well within tolerances in real-life applications. Relative errors of ground truth results for $f_0$ and $fbw$ are similarly small, with two $fbw$ exceptions for (1) and (6) due to simulator prediction errors. Thus, we expect the gap between model and ground-truth results to shrink as scaling grows. On the other hand, ground truth results show that the highest-reward solutions exceed specifications in $\max S_{21}$ for **5 out of 6 designs**. These examples demonstrate the ability of Alpha-RF to search for the optimal design for a variety of specifications.

**Comparison with Human Performance.** To further highlight the superiority of our design tool, we compare the six demonstrations above with **expert-level human designs** created by experienced RF engineers. As shown in Table 2, Alpha-RF achieves comparable or superior design performance

under the same specifications, while delivering solutions with a speedup of nearly **three orders of magnitude** compared to manual design processes.

Table 2: **Comparison with Human-Designed Filters.** Alpha-RF achieves comparable or superior rewards while significantly reducing design time.

| Design | Alpha-RF Reward | Human Reward | Alpha-RF Time (seconds) | Human Time (seconds) |
|--------|-----------------|--------------|-------------------------|----------------------|
| (1) | **0.8517** | 0.7466 | **7.3** | 9000 |
| (2) | **0.9783** | 0.9190 | **7.2** | 14760 |
| (3) | 0.9299 | **0.9781** | **7.4** | 14400 |
| (4) | **0.9813** | 0.8354 | **7.1** | 7200 |
| (5) | 0.9612 | **0.9699** | **7.3** | 8100 |
| (6) | 0.8484 | **0.9300** | **7.5** | 6300 |

## 3.3 ABLATION OF TEST-TIME SAMPLING.

We evaluate the impact of test-time sampling on final design quality. As shown in Figure 6, the mean reward steadily increases as the sampling budget grows, indicating that the sampling strategy itself directly improves performance as the budget increases. We also report the runtime scaling with sampling budget in Appendix D.

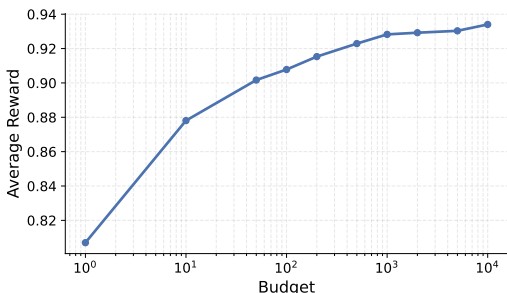

Figure 6: **Ablation of Test-Time Sampling.** Average reward as a function of the sampling budget. The curve shows that Test Time Sampling strategy improves performance as the budget increases.

## 4 LEARNING PHYSICS AND INTUITIONS

### 4.1 GENERALIZATION CAPABILITY

Although the neural simulator is trained to predict S-parameters of filter layouts defined by a fixed template, the simulator is trained not on template-based design parameters but on two-dimensional images of via footprint. Therefore, the model learns to generally predict S-parameters of a layout based on via placements on a two-dimensional grid. It is reasonable to assert that the model can simulate S-parameters of non-filter layouts constructed by vias. To verify this hypothesis, we use the neural simulator to predict the S-parameters of a different class of circuits with similar geometry but different frequency response, the waveguide (7). Unlike the filter, S-parameters of a waveguide are expected to show full transmission ($S_{21}(dB) \simeq 0$)) across the full frequency range. To construct these layouts, coupling openings are chosen to be $2.7$ mm, which is wider than the prescribed range of $[1.2, 2.6]$ mm. This allows microwave signals to pass through the circuit with minimal attenuation over the full frequency range of $26.5 - 40$ GHz. Predicted S-parameters of the test waveguide layouts are highly accurate compared to full-wave simulation results for different waveguide lengths, which are shown in Figure 7. These results demonstrate that our model exhibits generalization beyond our intended application (filter design).

### 4.2 LEARNING HUMAN INTUITION.

Through evaluation with many specifications, we discover that the automatic designer has adopted human design intuition. More specifically, the automatic designer is able to selectively target key design parameters for certain specifications to generate satisfactory designs, similar to a human

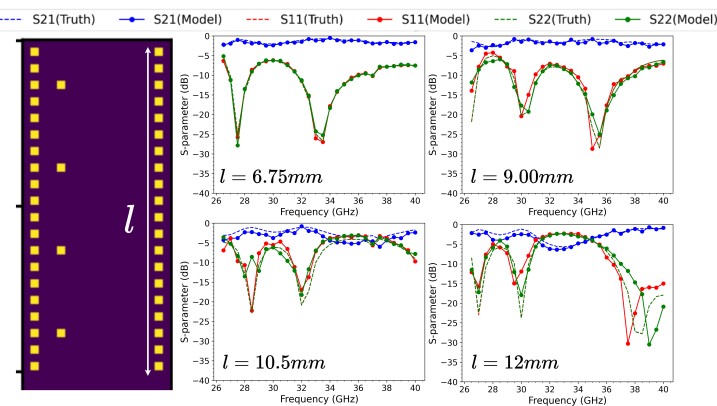

Figure 7: S-parameters for waveguide-like circuits of different lengths

designer. In the following paragraphs, we demonstrate this intuition for two specifications: center frequency $f_0$ and stop-band rejection $\alpha_r, \alpha_l$.

**Intuition for Center Frequency.** Center frequency $f_0$ is a function of the individual resonator length $L$ (5). Longer $l$ will result in lower $f_0$ and vice versa. To tune the center frequency of a design to specification, a human designer would primarily target resonator length. The same search intuition is seen by the agent when tasked to generate designs for $fbw = 0.07, \max S_{21} = -3, \alpha_r = -20, \alpha_l = -10$ and $f_0 \in [30, 32, 33, 35]$. $S_{21}$ of top solution for each variation is shown in Figure 8(a). As specified $f_0$ increases, $l$ of the best solution also decreases.

**Intuition for Stop-Band Attenuation.** Because stop-band attenuation is proportional to the number of resonators used (19), we expect the automatic designer to generate layouts with more sections for stricter $\alpha_r, \alpha_l$ (i.e higher stop-band attenuation), similar to how a human designer would determine value of $N$ through iterations. To verify this hypothesis, we generate designs for $f_0 = 33, fbw = 0.15, \max S_{21} = -1, \alpha_r \in [-15, -20, -30], \alpha_l = \alpha_r + 5$ and observe $S_{21}$. As specified $\alpha_r$ decreases from -15 to -20, the best solution increases number of resonators from 2 to 4 (Figure 8(b)).

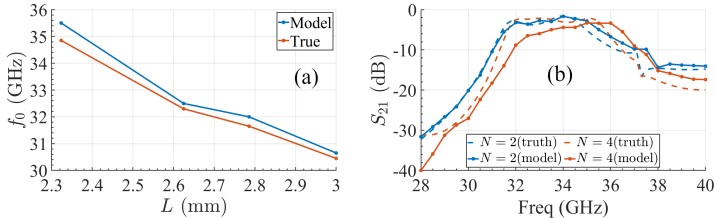

Figure 8: (a) Designed $l$ for different $f_0$ specifications (b) $S_{21}$ (dB) for designed $N$ for different $\alpha_r, \alpha_l$ specifications

## 5 DISCUSSION AND CONCLUSION

In this paper, we introduced Alpha-RF, an automated radio-frequency filter circuit design tool incorporating a scalable, fast, high-precision neural simulator to predict the S-parameters of filter layouts and an amortized inference policy leveraging the neural simulator to perform rapid, automatic optimization of the design. We demonstrated Alpha-RF through design examples where the tool generates optimized designs that perform as well as or, in some cases, exceed specifications. When compared with designs by experienced RF engineers, designs by Alpha-RF are comparable or superior, all the while reducing design time from hours to seconds. This is not surprising considering evidence of expert-like design intuitions. We also demonstrate the neural simulator's ability to generalized beyond filter designs, suggesting learning of the underlying physics. Given that general deep learning and reinforcement learning methods were used, we believe that our automatic design framework is transferrable to other domains and design problems in close affinity under different training examples, data representations and reward functions.

## 6 ETHICS STATEMENT

This work does not involve human subjects, sensitive data, or potentially harmful applications. Our methods are intended for scientific research purposes only.

## 7 REPRODUCIBILITY

All experiments presented in this paper are reproducible. The neural simulator architecture and reinforcement learning algorithm are described in details in the main paper and appendix.

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

# Appendix

## A   LLM Usage

LLMs were used only to polish language and improve writing efficiency. All research content is solely by the authors.

## B   Resonator-Coupled Band-pass Filter

### B.1   Construction

The resonator-coupled band-pass filter is built in a printed circuit board (PCB) with two metal layers and interconnecting vias as shown in Figure 9. In this construction, because clearance between adjacent vias is much smaller than the wavelength of the incoming RF signal, the signal is fully confined between the metal layers and two horizontal via rows across the length of the filter, forming a substrate integrated waveguide structure (7). Filtering response is realized by designing the amount of coupling between resonator sections, where coupling is controlled by dimensions $cw_0, cw_1, ...cw_7$. Resonator length $L$ determines the center frequency $f_0$ of the filter.

### B.2   S-Parameters

When an incoming RF signal travels inside the filter, we are interested in its transmission ("how much of the signal is transmitted?") and reflection ("how much of the signal is reflected back at the interface?") characteristics over frequency. The universal representation of frequency response for RF circuits is scattering parameters, or S-parameters (23). **At every frequency**, frequency response of the filter is characterized by the $2 \times 2$ scattering parameters (S-parameters) matrix

$$S = \begin{bmatrix} S_{11} & S_{12} \\ S_{21} & S_{22} \end{bmatrix}$$

where $S_{11}$, $S_{12} = S_{21}$[1], $S_{22}$ measure input reflection, transmission and output reflection, respectively. In most cases, we desire **low reflection** ($S_{11}, S_{22} \simeq 0$) and **high transmission** ($S_{21} \simeq 1$) within the **pass-band** frequencies, and the **inverse** in the **stop-band** frequencies ($S_{11}, S_{22} \simeq 1, S_{21} \simeq 0$). Examples of S-parameters are shown throughout section 3.

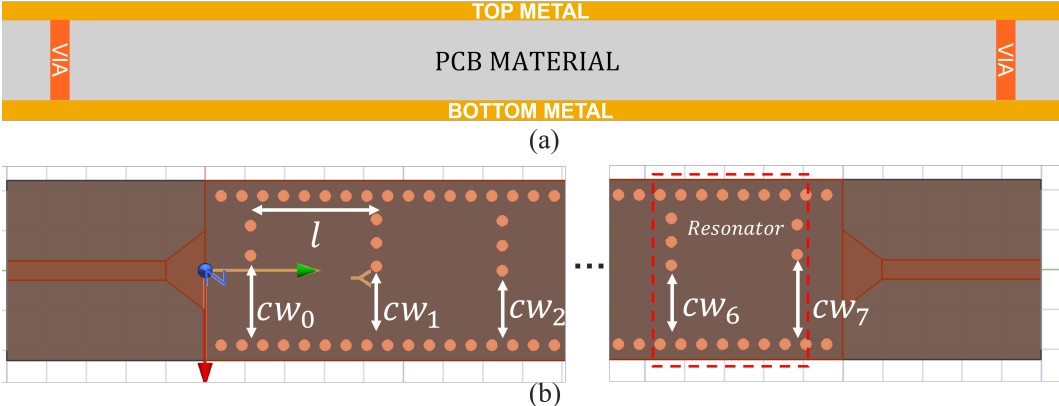

Figure 9: (a) Cross-section of the printed circuit board (b) Top-down view of the filter. Light orange circles are vias connecting the top and bottom metal, created by copper-plated drill holes. Rows and columns of vias create the filter structure, specifically the design parameters described in section 2.1

---

[1]This condition states that transmission is the same in both directions, which is always true for RF circuits without active elements e.g transistors

## C  ENVIRONMENT DETAILS

### C.1  SPECIFICATION RANGES

We list the ranges of target specifications used in the RL environment.

- **Center frequency** $f_0 \in [28.0, \ 39.0]$ GHz.
- **Relative bandwidth** fbw $= \frac{bw}{f_0} \in [0.02, \ 0.20]$.
- **Peak insertion loss** $\max S_{21} \in [-6.0, \ -1.0]$ dB.
- **Stop-band rejection** $\alpha_r, \alpha_l$ are $S_{21}$ at

$$f_l = 0.95\left(1 - \frac{\text{fbw}}{2}\right), \quad f_r = 1.05\left(1 + \frac{\text{fbw}}{2}\right),$$

with values in $[-60, \ -10]$ dB.

Range of $f_0$ is representative of our training dataset, which consists of filter layouts across $26.5 - 40$ GHz. Range of $fbw$ is typical for designs in real-life systems. Likewise, range of $\max S_{21}, \alpha_l, \alpha_r$ are considered practical and usable for real-life designs. Target specifications are sampled independently from uniform distributions over these ranges at each environment reset.

### C.2  REWARD FUNCTION CALCULATIONS

We reward the agent by how close the measurements of a candidate design are to specifications. To that end, we take the ratio between measurements and specifications to create continuous sub-rewards as follows

- $r_{f_0} = (\frac{f_0}{f_{0m}})^5$ if $f_0 < f_{0m}$, $r_{f_0} = (\frac{f_{0m}}{f_0})^5$ if $f_0 \geq f_{0m}$.
- $r_{fbw} = (\frac{fbw}{fbw_m})^3$ if $fbw < fbw_m$, $r_{fbw} = (\frac{fbw_m}{fbw})^3$ if $fbw \geq fbw_m$.
- $r_{maxS21} = \frac{maxS21}{maxS21_m}$ if $max_{S21m} \leq max_{S21}$, $r_{maxS21} = 1$ if $max_{S21m} > max_{S21}$.
- $r_{\alpha_r} = \frac{\alpha_{rm}}{\alpha_r}$ if $\alpha_{rm} \geq \alpha_r$, $r_{\alpha_r} = 1$ if $\alpha_{rm} < \alpha_r$.
- $r_{\alpha_l} = \frac{\alpha_{lm}}{\alpha_l}$ if $\alpha_{lm} \geq \alpha_l$, $r_{\alpha_l} = 1$ if $\alpha_{lm} < \alpha_l$.

To ensure the highest accuracy for $f_0$ and $fbw$ in the final design, $r_{f_0}, r_{fbw}$ are $5^{th}$- and $3^{th}$-order, respectively, to heavily penalize large differences in measurements and specifications. We also promote exploration of superhuman designs in calculations of $r_{\max S21}, r_{\alpha_r}, r_{\alpha_l}$ by maximizing these sub-rewards when specifications have been exceeded.

## D  RUNTIME VS. SAMPLING BUDGET

Table 3 reports the per-stage evaluation time for different sampling budgets. All evaluations were conducted on a single NVIDIA RTX 3090 GPU. As the sampling budget increases, the runtime growth is dominated by *Build Images* and *Neural-sim Forward*. The image-building stage is CPU-bound and cannot be fully vectorized, resulting in near-linear growth with batch size. For *Neural-sim Forward*, runtime increases more steeply at large budgets because GPU memory limits require splitting inference into multiple mini-batches rather than processing all samples in one pass, leading to additional overhead. These observations highlight where optimization efforts (e.g., parallelized image generation or memory-efficient model execution) could further reduce evaluation latency.

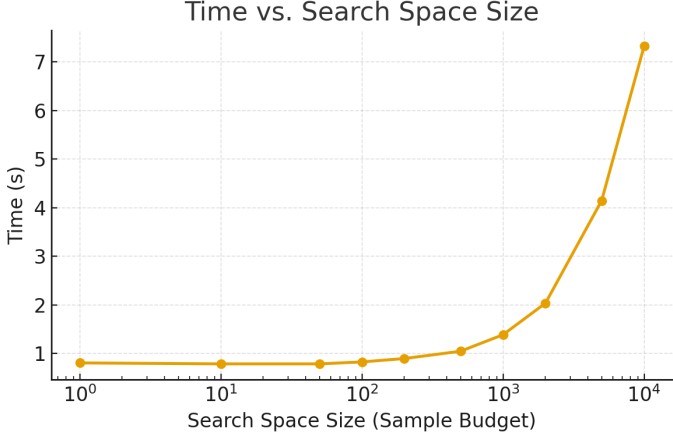

Figure 10: Runtime vs. sampling budget. As the budget grows, runtime increases roughly exponentially, mainly due to CPU-bound image building and batched neural-sim inference on GPU (see Appendix D for details).

Table 3: Per-stage evaluation time (seconds) for different sampling budgets. Total time includes policy sampling and all evaluation stages.

| Sample Budget | Policy Inference | Build Images | Neural-sim Forward | Interp |
|---|---|---|---|---|
| 1 | 0.35 | 0.18 | 0.27 | 0.00 |
| 10 | 0.35 | 0.17 | 0.26 | 0.00 |
| 50 | 0.35 | 0.19 | 0.24 | 0.00 |
| 100 | 0.36 | 0.21 | 0.25 | 0.00 |
| 200 | 0.35 | 0.26 | 0.27 | 0.01 |
| 500 | 0.34 | 0.38 | 0.31 | 0.01 |
| 1000 | 0.35 | 0.63 | 0.38 | 0.02 |
| 2000 | 0.35 | 1.08 | 0.56 | 0.04 |
| 5000 | 0.36 | 2.49 | 1.16 | 0.12 |
| 10000 | 0.36 | 4.59 | 2.15 | 0.21 |

