# OpenReview forum: "Alpha-RF: Automated RF-Filter-Circuit Design with Neural Simulator and Reinforcement Learning"
_ICLR.cc/2026/Conference — ICLR 2026 Conference Withdrawn Submission_

### Official Review · Reviewer_8SbP · 2025-10-16

**Soundness:** 2
**Presentation:** 2
**Contribution:** 2
**Rating:** 4
**Confidence:** 4

**Summary:**

This work demonstrates a neural network (CNN) trained on a large amount of labeled data to predict S-parameters. By leveraging this neural network, the authors trained another RL design agent to design RF filters. The experiment result show good accuracy of the model  and efficiency in terms of overall design time.

**Strengths:**

The RF filter design problem is an interesting problem.

**Weaknesses:**

1. Lack of original contribution
The neural network is based on a typical CNN
The RL training framework is based on a typical SAC framework
2. Lack of discussion and experiment comparison with existing RF filter design methods that also use ML approaches
As the authors point out, EM neural simulators have also been explored in RF integrated circuits (RFICs) design to rapidly predict frequency responses of RF circuits and structures, enabling automated design (15; 6). Is this work the first work targeting system design (RF filter specifically)?
3. Concerns regarding the method's generalization and applicability
The result only shows the generalization result to circuits with the same number of ports (two-port systems) and a fixed frequency band (26-40GHz). Lack of generalization results regarding circuits with more or fewer ports and different frequency bands.

**Questions:**

See weakness.

---

### Official Review · Reviewer_Z8Gy · 2025-10-17

**Soundness:** 3
**Presentation:** 3
**Contribution:** 1
**Rating:** 2
**Confidence:** 4

**Summary:**

Alpha-RF introduces a framework to automate the design of radio-frequency (RF) filter circuits, directly addressing the time-consuming and expertise-reliant nature of traditional design cycles that depend on slow electromagnetic (EM) simulations. The paper's core contribution is not merely predicting a circuit's performance but automatically generating an optimal design that meets specific performance goals—a paradigm shift from analysis to synthesis. The methodology is built upon two components: A CNN-based surrogate model (Neural Simulator) and Reinforcement Learning (RL) Agent It reduced a design cycle that typically takes days for an expert engineer to mere seconds, while producing designs that are comparable or even superior to human-level performance.

**Strengths:**

1. Alpha-RF applied AI framework to the notoriously difficult and practical domain of high-frequency RF circuit optimization.

2. Rather than merely predicting performance, the paper achieves true end-to-end design automation by ingeniously coupling an ultra-fast neural simulator with a reinforcement learning agent.

3. Achieved good results outperforming human experts.

**Weaknesses:**

While Alpha-RF demonstrates a significant achievement in automating a specific design task, its contributions are constrained by several notable weaknesses that limit its broader impact and generalizability.

**Lack of Geometric Generality:** The framework's primary limitation is its reliance on a single, fixed-topology template (a Substrate Integrated Waveguide). The system performs parametric optimization within this predefined structure rather than generating designs with true geometric or topological novelty. This template-based approach severely restricts its applicability, as it cannot address the vast space of arbitrary, multi-layer layouts common in real-world RF and PCB design, which graph-based methods are better suited to handle.

**Unproven Scalability:** The agent's success is demonstrated within a low-dimensional design space (one discrete and nine continuous parameters). The framework's viability for more complex problems remains questionable, as the performance of reinforcement learning agents often degrades significantly in high-dimensional continuous action spaces due to the curse of dimensionality, making exploration and credit assignment exponentially more difficult.

**Limited Algorithmic Novelty in AI:** From a machine learning perspective, the paper's contribution is more an application of existing methods than a development of new ones. The framework integrates a standard CNN architecture with a well-established continuous-control RL algorithm (TQC) to solve a problem that is structurally equivalent to a contextual bandit. While the integration is effective, the core AI components do not, in themselves, represent a fundamental advance in the field of machine learning.

**Insufficient Justification for Methodological Choices:** The paper lacks a rigorous ablation study to justify its choice of a complex reinforcement learning algorithm. Given that the problem is formulated as a single-step decision task (a contextual bandit), the necessity of a sequential RL framework like TQC/SAC is not self-evident. A thorough evaluation would have compared the chosen method against other powerful, non-sequential optimization techniques, such as evolutionary algorithms (e.g., GA) or Bayesian Optimization.

**Questions:**

I have the following questions:

1.  It is difficult to understand the rationale for exclusively using a Reinforcement Learning (RL) algorithm for what appears to be a single-step optimization problem. Could you elaborate on why other methods, such as Bayesian Optimization (BO), were not considered or implemented?

2.  My understanding of the proposed methodology is that the neural simulator is used for the optimization process, and the real simulator is only engaged for the final evaluation. Could you please confirm if this interpretation is correct?

3.  Reference [1] also takes S-parameters into account and seems to address a more general case, even though it does not extend to full optimization. From the authors' perspective, are there significant technical challenges that make it difficult to apply the more generalized approach of [1] to the optimization task in your work?

[1] Kim, Doyun, et al. "TraceFormer: s-parameter prediction framework for PCB traces based on graph transformer." Proceedings of the 61st ACM/IEEE Design Automation Conference. 2024.

---

### Official Review · Reviewer_7Eto · 2025-10-28

**Soundness:** 2
**Presentation:** 3
**Contribution:** 1
**Rating:** 2
**Confidence:** 5

**Summary:**

The paper proposes to use a neural surrogate model + reinforcement learning agent to achieve efficient design of analog filters.
Experiments show that the toolkit enables significant design efficiency over human designers, yet nearly equivalent design quality as human designs.
The neural surrogate model can also be transferred to predict the performance of other circuits, i.e., waveguides.

**Strengths:**

+The key formulation flow of the proposed methods is presented well.

+Experimental results on the benchmark circuit are also demonstrated well.

**Weaknesses:**

-The entire design flow is not new, basically following the same methods presented by many prior work, although prior work does not exactly design an RF filter.

[1] 25.3 AI-Enabled Design Space Discovery and End-to-End Synthesis for RFICs with Reinforcement Learning and Inverse Methods Demonstrating mm-Wave/sub-THz PAs Between 30 and 120GHz;
[2] Deep Learning Aided Modelling and Inverse Design for Multi-Port Antennas;
[3] INSIGHT: A Universal Neural Simulator Framework for Analog Circuits with Autoregressive Transformers;
[4] AdreamDCO: AI-Driven Robust and Efficient Design Automation for Digitally Controlled Oscillators.

-While the work shows that the surrogate model can be applied to waveguides, this is not a model-level innovation, pretty much due to the similarity of geometry between RF filters and waveguides. The design flow is quite circuit-specific, which cannot generalize well to multiple circuits. Overall, there is limited domain-specific adaptation of using these learning-based methods.

-Lots of technical details need justifications. For example, why is a CNN-based surrogate model is used, not others? Why is soft-award used for RL agent design? Why not directly search optimal design instead of rolling out (i.e., doing inference) many times? What is the exact problem definition of the filter design? An optimization problem, right? What is the data size used to train the RL agent?

-Given that lots of similar methods have been proposed in the domain, a thorough comparison is required.

**Questions:**

For the neural surrogate, the number of inputs is limited, but why can it predict the circuit specifications in a dimension of 128 that is significantly beyond the size of the input?

**Details Of Ethics Concerns:**

N/A.

---

### Official Review · Reviewer_doJV · 2025-10-31

**Soundness:** 3
**Presentation:** 3
**Contribution:** 2
**Rating:** 4
**Confidence:** 4

**Summary:**

The authors propose a new method for the automatic design of RF circuits via a neural simulator combined with an RL agent. First, the neural simulator is trained using an existing simulator. Second, the authors train an RL agent given a target specification to predict RF circuit parameters.

**Strengths:**

1. Efficient surrogate modeling of expensive physics simulations.
2. Novel application of RL to a real engineering problem.
3. Reward design aligns with human engineering priorities.

**Weaknesses:**

While the authors claim that the neural simulator generalizes beyond the filter layout template because it operates on 2D via images, the evidence provided is not convincing. The “waveguide” test case still follows the same template topology, differing only by a slightly widened coupling gap. This tests mild extrapolation within the same geometric manifold, rather than genuine generalization to unseen circuit topologies or layout patterns. Demonstrating true generalization would require evaluation on layouts with different topologies or boundary conditions, not merely parameter values outside the training range.

**Questions:**

1. It feels that, instead of MAE, MAPE would be more preferable to show the error.
2. How was human time estimated?
3. Could you clarify how the distance is encoded? Is it something like - distance between vias in the image = sampled cw_i converted from mm → pixels.
4. How often does the simulator itself fail? Are there any issues with that?
5. What happens in Figure 5 (4) for S22? Does the simulator here have a huge error?
6. Can you provide a compelling argument for generalization besides slightly tuning one of the parameters? I.e can we generalize on other templates?

---

### Note · Authors · 2025-11-14

I have read and agree with the venue's withdrawal policy on behalf of myself and my co-authors.